# Liquid Silicone Rubber Foamed with Thermoplastic Expandable Microspheres

**DOI:** 10.3390/ma15113779

**Published:** 2022-05-25

**Authors:** Svenja Marl, Ralf-Urs Giesen, Hans-Peter Heim

**Affiliations:** Institut Für Werkstofftechnik—Kunststofftechnik, University of Kassel, 34125 Kassel, Germany; giesen@uni-kassel.de (R.-U.G.); heim@uni-kassel.de (H.-P.H.)

**Keywords:** liquid silicone rubber, LSR, foams, injection molding, thermoplastic expandable microspheres

## Abstract

To reduce the material costs as well as the density of Liquid Silicone Rubber (LSR), LSR foams can be produced in an injection molding process. Expandable thermoplastic microspheres can be used as blowing agents. This publication deals with the analysis of the cell structure of these LSR foams. For this purpose, cylindrical disks are injection molded and examined for their cell structure as a function of different proportions of microspheres using a scanning electron spectroscope. In addition, the density of the samples is determined. It was found that a very homogeneous cell structure is produced in this process, that heat transport has a significant influence on the expansion of the microspheres, and that the formation of a filler network limits the expansion at higher proportions of blowing agent.

## 1. Introduction

In plastics technology, the processing of silicone rubber is becoming more significant. Silicone rubber is given preference over other elastomers because of its good processing ability, temperature resistance up to 300 °C, physiological compatibility, high transparency, and resistance against a lot of chemicals and environmental impacts. Due to these properties, this material class will often be used in the automotive, medical, household and energy sector.

In all above-mentioned fields, the use of foamed silicone parts is possible and affects the components by making them softer, cheaper and lighter. Foamed thermoplastic parts have been used for a long time in the market. As a further goal, next to the weight reduction of the components, the foaming process has a positive effect to the injection molding with regard to mold filling and the avoidance of residual stresses. Foamed components are produced by injection molding, press molding and extrusion. Examples are shoe soles, sealings, instrument panel bases, grips, and pipe insulation [1,2,3].

Known processes for foaming silicone rubbers [4] are the processes of physical and chemical foaming which are also used in the processing of thermoplastics [5]. Chemical foaming requires organic or inorganic compounds added to the silicone rubber. Parallel to the vulcanization process of silicone rubber, nitrogen or carbon oxides are formed from the additives at high temperatures. Furthermore, they become free and expand, creating foam sections [6]. During this foam production, a result could be toxic substances that emit directly or later from the components. In physical foaming processes, gases such as nitrogen or carbon dioxide are added to the thermoplastic melt in the injection molding or extrusion unit under high pressure. In addition, there is a process by Arburg in which the granulate is first loaded with gas, usually nitrogen and carbon dioxide, in the autoclave [7]. If the gas-loaded melt comes into an empty cavity, the pressure drops. The gas expands and foam will be produced. First experiments with physical foaming processes for silicone rubber in injection molding were carried out in the beginning of the 2000s [8]. Negative effects were observed in process control; also, reproducible production of components was not possible [9]. 

Métivier and Cassagnau summarize very well different manufacturing methods and foaming processes of silicone rubber in [10]. However, the process illustrated here differs from the processes described there because, depending on the process, pre-crosslinking is dispensed with, or, in the case of the synthetic foams mentioned, the microspheres are already pre-expanded or made of glass.

Using expandable thermoplastic microspheres to foam silicone rubbers can be a solution to produce silicone elastomer foam [3]. The used microspheres consist of a thermoplastic shell filled with gas (e.g., isopentane). Heating the shell in the hot mold cavity softens it and the foaming gas causes the microsphere to expand (Figure 1). The microspheres are added to the silicone rubber in proportions of up to 3% by weight before processing. In the investigations, it will be shown how the density will change and how the cell distribution and size are in the material. The mechanical properties of silicone foam with thermoplastic expandable microspheres are explained in detailed in [11]. 

The most important results from [11] for the material combination investigated in this paper are that the density of the tensile test bar with 3 wt-% microspheres can be reduced by up to 47%. The density reduction depends on the position in the specimen. Together with the density reduction and the resulting lower matrix content, the tensile strength also decreases by 67% at 3 wt-% microspheres and the elongation at break by 12.4% (Figure 2). 

Since the specific tensile strength also decreases by up to 45% at 3 wt-% blowing agent, it can be concluded that there is no adhesion between the microspheres and the matrix. It could also be determined there by the change in hardness and stress at 100% elongation that these are significantly changed by the hardness of the shell material of the microspheres [11].

The difference between this liquid injection molding (LIM) of silicone rubber and the foaming of thermoplastic injection molding is the temperature control. Whereas in thermoplastics the screw is heated, and the mold is cooled, in LIM the screw is cooled, and the mold is heated. This effect is used here to expand the blowing agent in parallel with the crosslinking of the LSR in the hot cavity. In the case of thermoplastic elastomers (TPE), microspheres are already being used commercially in the injection molding process, since process control is significantly simpler here due to the absence of crosslinking and the different temperature control [12]. 

## 2. Materials and Methods

The used liquid silicone rubber was the QP1-30 of Dow Chemicals. This is a two-component addition-cured silicone rubber with a low viscosity and can be used for medical devices. 

As blowing agent, the thermoplastic expandable microspheres Expancel 031 DU 40 by Nouryon (Sundsvall, Sweden) were used. These microspheres start the expansion between 80 and 95 °C. The diameter of the unexpanded microspheres is between 10 and 16 µm. The microspheres used here are those with the lowest expansion temperature. Experiments with other microspheres that expand at higher temperatures has shown that the crosslinking of the LSR occurs faster than expansion, resulting in a much smaller reduction in density reductions. Therefore, a comparison between different types is difficult or not possible.

The amount of blowing agents were 1 wt-%, 2 wt-% and 3 wt-%. For the mixing process the hand mixer Grundig HM 7680 was used to mix both components of the LSR with the expandable microspheres in the cartridge. This cartridge is produced by Fischbach KG (Engelskirchen, Germany) and has a volume of 500 mL. The volume of 500 mL is given by the dosing system of the injection molding machine (see in Figure 3). However, tests with other dosing systems on a different injection molding machine showed that there is no influence. Due to the direct further processing after mixing, it can be assumed that there is no segregation or aggregation of the microspheres in the LSR. Previous studies with a 200 ct silicone oil have shown that the mixed-in microspheres did not separate over a period of several days, which is why this has not yet been investigated further in the LSR. 

For these tests, cylindrical disks with a height of 8 mm and a diameter of 29 mm were produced by injection molding. To produce these disks, the injection-molding machine Babyplast 6/10P from Christmann Kunststofftechnik was used. This injection-molding machine has a cartridge dosing system and a piston injection unit. The mold temperature was 160 °C for all samples and the injection volume depends on the amount of blowing agent. Previous tests [3] have shown that with this combination of materials, the lowest densities could be achieved at 160 °C. At lower temperatures, crosslinking is not fully completed after two minutes, which means that post-expansion takes place after demolding and the components are not dimensionally stable. A longer crosslinking time than two minutes cannot be set on this injection molding machine. A higher mold temperature, on the other hand, leads to faster crosslinking of the LSR, which hinders the expansion of the microspheres.

The injection volume (see Table 1) was determined by a filling study where the value at which the mold was just filled was selected. The vulcanization time varies with increasing amount of blowing agent (see Table 1), since more gas is present in the test specimen due to the higher proportion and the gas phase impedes the heat transport and thus the crosslinking of the silicone. As a result, the vulcanization time increases as the proportion of blowing agent increases.

The density was determined using a YDK03 density determination set from Sartorius (Goettingen, Germany). Water was used as the medium. For the non-foamed reference samples, the immersion basket was used because the density is greater than 1 g/cm^3^. For the foamed samples, the immersion sieve had to be used, as they are lighter than 1 g/cm^3^. 3 samples were tested in each case.

In addition, scanning electron microscopy (SEM) images were taken to investigate the cell distribution of the samples. For this purpose, the SEM CamScan MV2000 from Electron Optic Services Inc. (Ottawa, ON, Canada) was used with a SE detector. The specimens were cut midway through the injection point and the cross-sectional area is analyzed for the distribution of microspheres at 5 positions (Figure 4). The surface was sputtered with gold. The magnification is 250× and the accelerating voltage was selected as 10 kV. 

The cell size or cross-sectional area of the microspheres was determined by manually tracing the cells using ImageJ software and measuring the area. These values could afterwards be evaluated using Excel. For the sample cross-section examined, it can be assumed that the individual microspheres were not cut in their centers. This leads to a statistical error in the results. However, since this error occurs in all samples examined, it will not be further investigated.

In order to understand the mold filling and thus the heat transfer from the mold wall into the material, the mold filling was simulated using SigmaSoft Virtual Molding (version 5.3.0.6, simulation created by Kevin Klier, SIGMA Engineering GmbH, Aachen, Germany). Figure 5 shows the unfoamed mold filling of 60% for 3 wt-% (see Table 1).

## 3. Results

### 3.1. Density

The reference disks have a density of 1.12 g/cm^3^. After adding 1 wt-% of microspheres the density decreases to 0.786 g/cm^3^, which is a decrease of 30%. With 2 wt-% of microspheres the density decreases up to 45% compared with the reference to 0.618 g/cm^3^. The lowest density has the sample with 3 wt-% of blowing agent with 0.524 g/cm^3^. In comparison to the reference is this a decrease of 53%. The density is shown in Figure 6.

### 3.2. SEM Pictures

The cell distributions are shown in the following figures. In Figure 7, the cell distributions in the center of the cross-section (position 1) are shown as a function of the proportion of microspheres. The proportion of silicone matrix decreases with an increasing proportion of microspheres. In Figure 8, the distribution as a function of the positions using the example of 2 wt-% microspheres. The distribution of the microspheres within the sample is very homogeneous in this process. This makes this foaming process very different from other processes, which often have an inhomogeneous foam structure. In addition, it can be seen in the images that the structure is closed-cell.

### 3.3. Cellstructure

The cell structure was analyzed by the area of the microspheres in relation to the total area as a function of the positions (see Figure 4 and Figure 9). With increasing proportion of blowing agent, the proportion of microspheres increases for each position. The increase is logarithmic, flattening out as the proportion of microspheres increases. Between the different positions it can be observed that in the middle of the sample (Pos. 1) at 1 wt-% the highest matrix content (lowest proportion of microspheres in the total area) is present. At 2 wt-% and 3 wt-%, the degree of foaming is thus highest in the center of the sample. Irrespective of the blowing agent content, the highest proportion of microspheres per unit area can always be observed in Pos. 4, i.e., in the near vicinity of the injection point. Pos. 2 shows only slightly lower values. With the exception of 1 wt-%, the lowest area fractions of microspheres can be observed in Pos. 3 and Pos. 5. 

Figure 10 shows the average cross-sectional area of the microspheres as a function of the proportion of microspheres over the various positions. If positions 2 to 5 at 1 wt-% are compared with the same positions at 2 and 3 wt-%, it is noticeable that the larger microspheres are found at 1 wt-%. In combination with the results from Figure 9, that at 1 wt-% the highest proportion of matrix is present, it can be concluded that the number of microspheres is lower, but they are proportionally larger than at higher proportion of blowing agent. The largest cross-sectional areas can thus be observed at the lowest proportion of blowing agent, and as the proportion of microsphere increases, the size decreases regardless of position. 

The mean value of the area of the microspheres over the individual positions for all proportions is shown in Figure 11. The dashed line indicates the mean value over all microspheres of a blowing agent proportion, which again shows that the size of the microspheres in the entire sample decreases with increasing proportion of blowing agent. Here it is more obvious that positions 2 and 4 have the larger microspheres than positions 3 and 5. 

## 4. Discussion

Based on the results, three conclusions can be drawn from these experiments. The first conclusion is that the size of the microspheres, determined here on the basis of the cross-sectional area, decreases as the proportion of microspheres increases. This means that the microspheres hinder each other from expanding, which also explains the exponential decrease of the density curve. Since the microspheres influence each other, it can also be concluded that a filler network is formed between one and two percent by weight. In addition, the heat transport through the silicone rubber also influences the crosslinking as well as the expansion. The addition of the microspheres introduces the blowing gas into the microspheres. Since gaseous materials have a lower thermal conductivity than plastics, heat transport is inhibited by the smaller amount of matrix as the proportion of blowing agent increases. This also elaborates the higher crosslinking time at 3 wt-% (see Table 1).

The second conclusion is that early contact with the mold wall leads to larger microspheres, since in Pos. 4 and Pos. 2 there is direct contact between the mold wall and the material after the injection process (see Figure 4 and Figure 5). Heat transport takes place directly, allowing the microspheres to soften and expand first. This process occurs in parallel with the crosslinking of the LSR, inhibiting expansion later in the process. In Pos. 3 and Pos. 5, there is an air cushion between the mold wall and the material at the beginning, which restricts heat transport and limits the speed at which the microspheres can expand. Thus, the main heat flow through the silicone matrix comes from the other side of the cavity, which means that the expansion starts here the latest and has a higher backpressure due to the progressive crosslinking and associated viscosity increase. Here it is interesting to investigate further mold geometries in the future to allow different directions of expansion. For example, a central injection point for the cylindrical disk would allow a different mold filling compared to Figure 5, whereby a more uniform expansion in all directions can be expected with a sub volumetric mold filling. The tensile test bar, as studied in [11], has a much smaller cross-section with a longer flow length. Density tests have shown that the density is lowest in the area furthest away from the injection point. Due to the small cross-section, there is better heat transport through the LSR, unlike in the studied disk, and thus a longer expansion time. The investigation of the cell distribution over different positions along the flow path is still pending, but the density is already providing initial insights into this. 

The third conclusion is the very homogeneous cell structure over the entire sample. Compared to other foaming methods, such as water foamed LSR samples in [13], the size of the pores at different positions is very similar. The homogeneous cell structure results from the nature of the microspheres. Due to the thermoplastic shell enclosing the foaming gas, collapse of multiple cells is not possible. With other blowing agents such as water (cf. [13]), multiple nucleation centers can grow together, resulting in very inhomogeneous cell structures. In the case of microspheres, on the other hand, the shells touch each other in the process, but the gas remains separated. This results in a filler network. In addition, the maximum size of the microspheres is limited because the foaming gas in the shell can only build up a certain pressure so that the shell expands. Expansion is complete as soon as the internal and external pressures are equal. Possible influencing variables on the external pressure are the evacuation of air from the mold and the viscosity of the LSR. Since the viscosity depends on the degree of crosslinking and increases as the degree of crosslinking increases, the external pressure is increased during crosslinking and expansion is inhibited. Thus, the maximum size of the microspheres in the LSR is limited. Further studies show a homogenous cell structure for LSR-foams with microspheres in the cross-section area of other geometries like the tensile bar S2 in [11]. When comparing the structure of injection molded TPE foams foamed with microspheres with these samples, the TPE foams do not contain a continuous foam but a compact edge layer [12]. One possible reason for this difference is the different temperature control in the injection molding process. The microspheres do not have the opportunity to expand on the cold mold wall (thermoplastic injection molding), which means that the solidified surface layer remains unfoamed and only expands in the hot melt in the center of the mold.

## 5. Conclusions

In summary, density reductions of up to 53% were achieved with microspheres. The cell structure is very homogeneous. In addition, heat transport has a significant influence on the expansion of the microspheres, and the formation of a filler network can also limit the expansion. 

## 6. Patents

EP000003331682B1.

## Figures and Tables

**Figure 1 materials-15-03779-f001:**
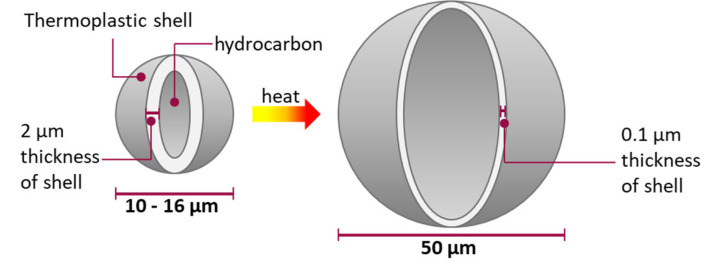
Concept of expansion of thermoplastic microspheres. Ref. [11] CC BY-SA 4.0 international.

**Figure 2 materials-15-03779-f002:**
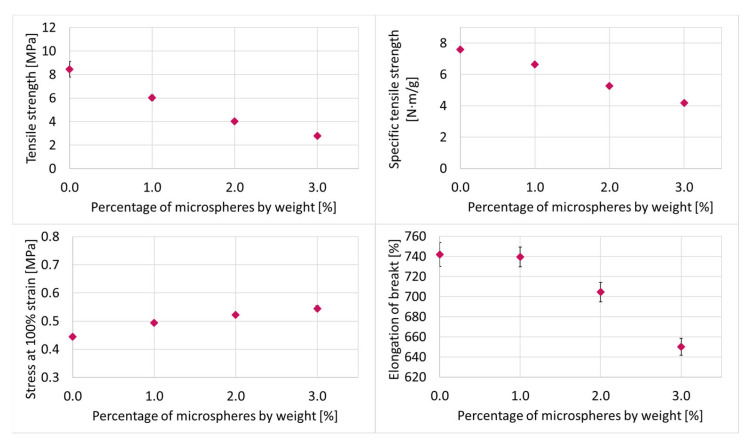
Results of the tensile test of QP1-30 with Expancel 031 DU 40 [11].

**Figure 3 materials-15-03779-f003:**
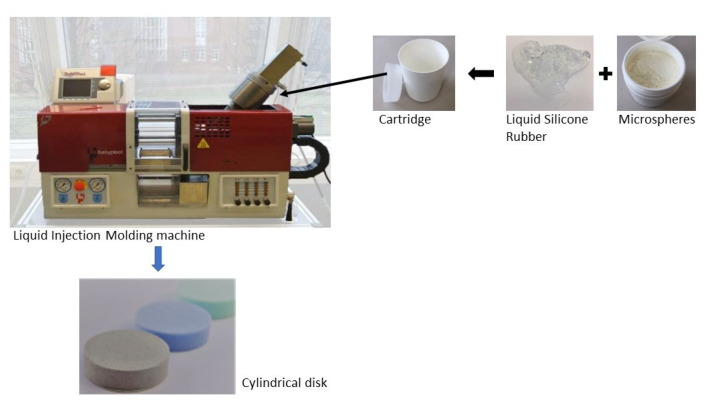
Liquid Injection Molding (LIM) machine to produce foamed LSR with thermoplastic expandable microspheres.

**Figure 4 materials-15-03779-f004:**
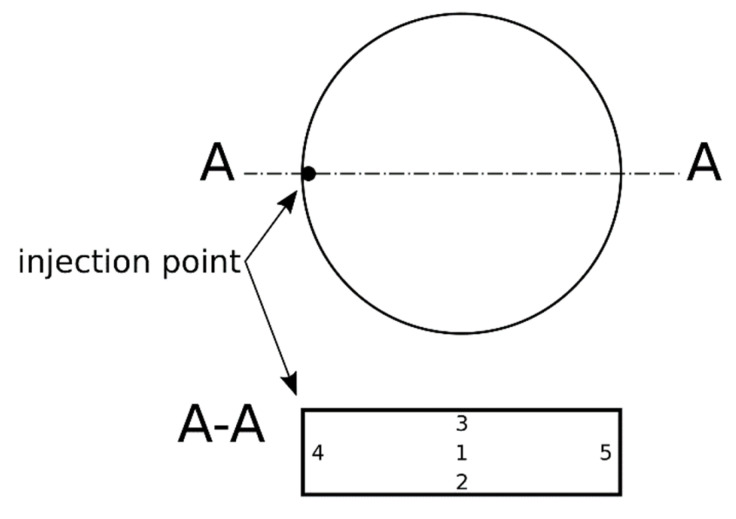
Specimen preparation and numbering of positions depending on the injection point.

**Figure 5 materials-15-03779-f005:**
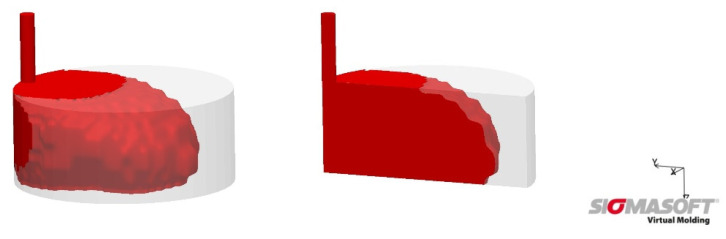
Simulation of mold filling for 60%, corresponds to sub volumetric mold filling for 3 wt-% without expansion of the microspheres.

**Figure 6 materials-15-03779-f006:**
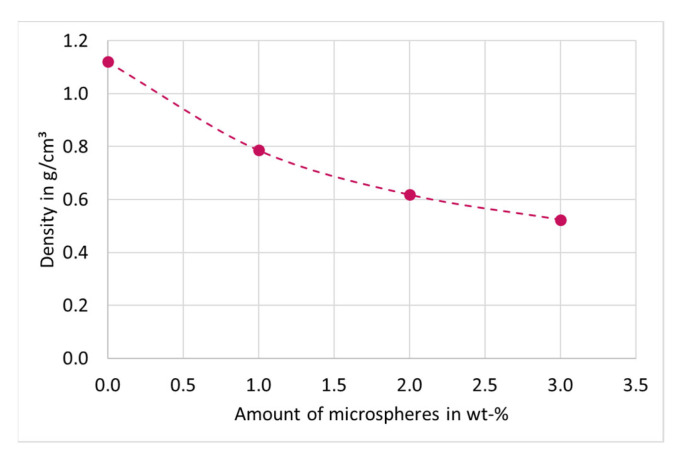
Reduction of the density depending on the amount of microspheres. The dashed line does not correspond to a fitting curve, but is for optical purposes.

**Figure 7 materials-15-03779-f007:**
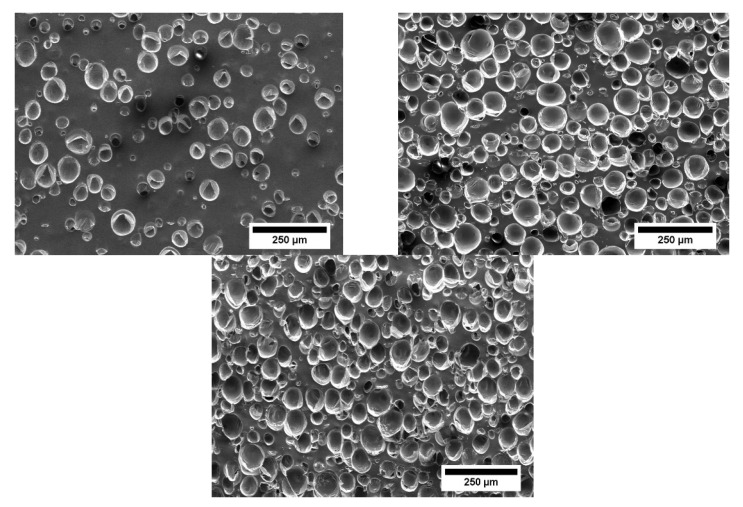
SEM images for position 1 at different proportions of microspheres (**top left**: 1 wt-%; **top right**: 2 wt-%; **bottom** 3 wt-%).

**Figure 8 materials-15-03779-f008:**
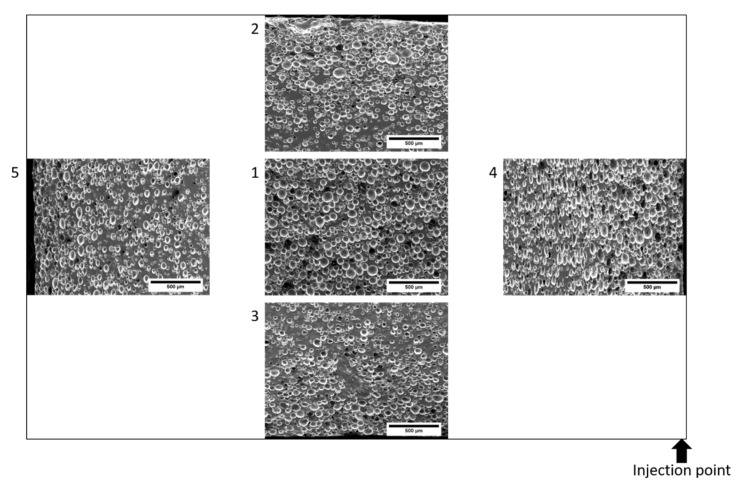
SEM images as a function of positions to the injection point for 2 wt-% microspheres.

**Figure 9 materials-15-03779-f009:**
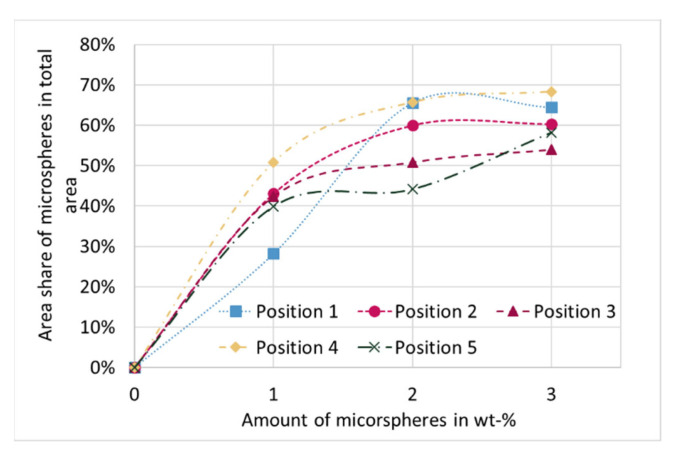
Proportion of microspheres to total area for different proportions of microspheres as a function of positions. The dashed lines do not correspond to a fitting curve but are for optical purposes.

**Figure 10 materials-15-03779-f010:**
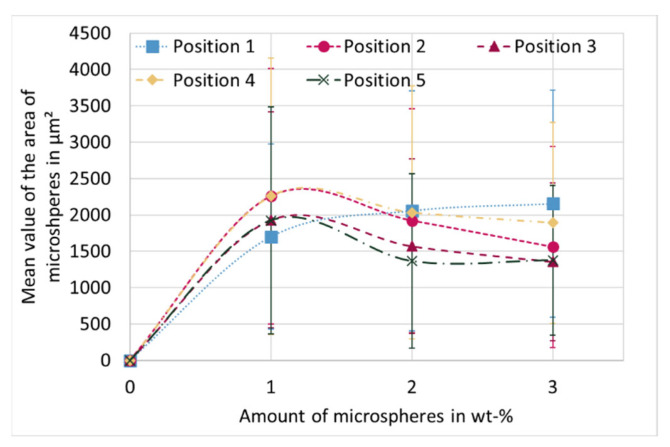
Averaged cross-sectional area of microspheres for different proportions of microspheres as a function of positions. The dashed lines do not correspond to a fitting curve, but is for optical purposes.

**Figure 11 materials-15-03779-f011:**
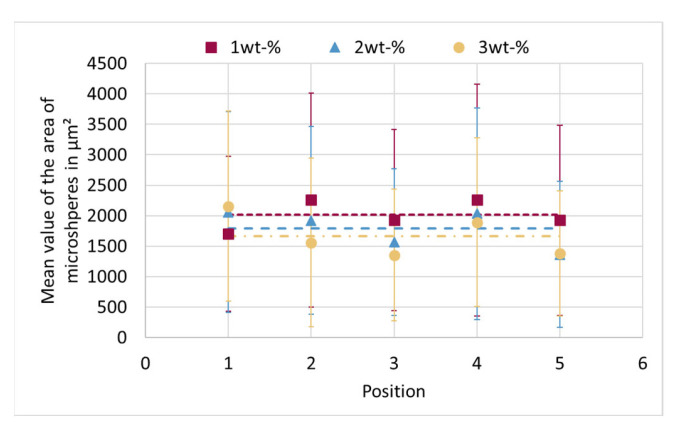
Averaged cross-sectional area of microspheres for different positions as a function of proportions of microspheres. The dashed lines indicate the averaged area of the microspheres over all positions. The dashed lines do not correspond to a fitting curve but are for optical purposes.

**Table 1 materials-15-03779-t001:** Manufacturing parameters depending on the blowing agent content.

Amount of Blowing Agent in wt-%	Injection Volume in		Vulcanization Time in s
	cm^3^	%	
0	12.5	100	80
1	9.5	76.0	110
2	8.1	64.8	110
3	7.6	60.8	120

## Data Availability

Not applicable.

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
