# Peer review of "Liquid Silicone Rubber Foamed with Thermoplastic Expandable Microspheres"

_materials, 2022, doi:10.3390/ma15113779_

Round 1
Reviewer 1 Report
The manuscript of “Liquid Silicone Rubber foamed with thermoplastic expandable Microspheres” had been reviewed. However, it is shown that it seems to be a experiment report, not a research. The variables are very limited only including the loading of the blowing agent, and the injection positions. Actually, different kinds of blowing agents with different initial expansion temperature and curing temperature are also important to affect the foaming performance, which has not been studied. Moreover, the cartridge used for the foaming is 500 mL, we thick that if we choose a larger or smaller container and vary the injection positions, the conclusions might be changed. Thus, the manuscript needs to be designed again to further discover the foaming behaviors of the LSR.
Author Response
Dear Reviewer,
Thank you very much for your constructive criticism. It helped us a lot to understand which parts of the text were not described in enough detail. For example, we have described the process concerning the cartridge system in more detail and shown it graphically.
In addition, we now elaborate more on the defined parameters (mold temperature, selected microspheres) and explain why these were selected.
To further explain the processes and results, we now refer to additional literature. Sources 10, 11, and 12 have been added.
Yours sincerely,
Svenja Marl
Reviewer 2 Report
This work presents an approach to produce foamed silicone rubber by adding expandable microspheres during injection molding process. Although the work seems well-written, it show little novelty to the academic society as most of the data shown in this manuscript are predictable. Hence, I am unable to recommend for publication.
Other comments:
- Since the authors mentioned other blowing techniques, it would be better to perform these techniques as comparison.
- The dispersion of expandable microspheres in LSR is not well characterized. Will they aggregate or sedimentate over time or in bulk sample?
- The authors only shows the data with the LSR shape of cylindrical disks. However, uniformed foaming in irregular molding shape is demanded and difficult to achieve in reality.
- Fig.3 only contains 4 data points. A fitting curve with these four data makes no sense.
Author Response
Dear reviewer,
Thank you very much for your constructive criticism. It helped us a lot to understand which parts of the text were not described in enough detail.
For further explanation of the processes and results, we now refer to additional literature. Sources 10, 11, and 12 have been added. It was also briefly explained what the differences are with the current innovations and applications of silicone foams, which were very well summarized in source 10 (10. Métivier, T.; Cassagnau, P. Journal of Cellular Plastsics 2019, 55 (2), 151-200 https://doi.org/10.1177%2F0021955X18806845).
Through source 11, we now explain how other mold geometries can lead to other cellular structures. (11. Marl, S.; Hartung, M.; Giesen, R.-U.; Heim, H.-P. Proceedings SPE FOAMS® 2019 Conference, Valladolid, Spain (2-3 October 2019) http://dx.doi.org/doi:10.17170/kobra-202204266098)
The third source allows us to understand the difference in cell structure between TPE foamed with microspheres and LSR (12. Ries, S.; Spoerrer, A.; Altstaedt, V. AIP Conference Proceedings 2014, 1593, 401-410 https://doi.org/10.1063/1.4873809).
We would be pleased if we can now convince you of the publication.
Yours sincerely,
Svenja Marl
Reviewer 3 Report
Dear Authors,
This paper represents the formation of silicone rubber foams with thermoplastic expandable microsphere as new type blowing agent. The data are well presented. However I have a curiosity about the physical properties of those foams. Authors should have to provide at least compressive mechanical properties for the value of industrial application.
Author Response
Dear reviewer,
Thank you very much for the positive review. We have now referred you to other sources for more detailed information on the mechanical properties. Especially source 11 (http://dx.doi.org/doi:10.17170/kobra-202204266098) describes the mechanical properties in great detail.
Yours sincerely,
Svenja Marl
Round 2
Reviewer 1 Report
The authors have revised the manuscript according to the comments, and I think it can be accepted.
Author Response
Dear Reviewer,
Thank you very much for your expert opinion.
Due to one of the other reports, we have added two more paragraphs to the introduction, which is why there has been another round of review.
Yours sincerely,
Svenja Marl
Reviewer 2 Report
Although the authors have placed some additional reference and explanation in the revised version, to be honest, I still think more experiments should be conducted to support the authors' concept as well as to prove its novelty. However, I understand the authors may not want to put extra effort on the lab work to further enhance the quality of this work. Considering the manuscript has been improved compared to the previous version, I think it can be published.
Author Response
Dear Reviewer,
Thank you very much for your expert opinion. Additional work in the laboratory was unfortunately not possible in this case due to the short revision time (10 days or 2 days) given by MDPI. Therefore, we have chosen the method of additional explanations and references to other literature.
I will try to implement your constructive criticism directly in future publications. Again, thank you for taking the time to review the publication.
Yours sincerely,
Svenja Marl
Reviewer 3 Report
Dear Author,
I apreciate your revision based on my suggestion. However, I think only mentioning the reference is not sufficient. Therfore, I recomend to add sumarized mechanical properties in the introtuction so that reader can get an idea.
Author Response
Dear reviewer,
Thank you very much for your expert opinion. As we have received from the editor again the opportunity to revise, we have still added the mechanical properties of the mentioned source in the introduction.
Yours sincerely,
Svenja Marl